# Dental Implant Surface Decontamination and Surface Change of an Electrolytic Method versus Mechanical Approaches: A Pilot In Vitro Study

**DOI:** 10.3390/jcm12041703

**Published:** 2023-02-20

**Authors:** Mariana Anselmo Assunção, João Botelho, Vanessa Machado, Luís Proença, António P. A. Matos, José João Mendes, Lucinda J. Bessa, Nuno Taveira, Alexandre Santos

**Affiliations:** 1Egas Moniz—School of Health and Science, 2829-511 Almada, Portugal; 2Egas Moniz Center for Interdisciplinary Research, Egas Moniz—School of Health and Science, 2829-511 Almada, Portugal

**Keywords:** peri-implantitis, implants, decontamination systems, *Pseudomonas aeruginosa*, electrolytic decontamination, dental implants

## Abstract

Dental implants are the preferred fixed oral rehabilitation for replacing lost teeth. When peri-implant tissues become inflamed, the removal of plaque accumulating around the implant becomes imperative. Recently, several new strategies have been developed for this purpose, with electrolytic decontamination showing increased potential compared to traditional mechanical strategies. In this in vitro pilot study, we compare the efficacy of an electrolytic decontaminant (Galvosurge^®^) with an erythritol jet system (PerioFlow^®^) and two titanium brushes (R-Brush™ and i-Brush™) in removing *Pseudomonas aeruginosa* PAO1 biofilms from implants. Changes in the implant surface after each approach were also evaluated. Twenty titanium SLA implants were inoculated with *P. aeruginosa* and then randomly assigned to each treatment group. After treatment, decontamination efficacy was assessed by quantifying colony-forming units (log10 CFU/cm^2^) from each implant surface. Scanning electron microscopy was used to analyse changes in the implant surface. With the exception of R-Brush, all treatment strategies were similarly effective in removing *P. aeruginosa* from implants. Major surface changes were observed only in implants treated with titanium brushes. In conclusion, this pilot study suggests that electrolytic decontamination, erythritol-chlorhexidine particle jet system and i-Brush™ brushing have similar performance in removing *P. aeruginosa* biofilm from dental implants. Further studies are needed to evaluate the removal of more complex biofilms. Titanium brushes caused significant changes to the implant surface, the effects of which need to be evaluated.

## 1. Introduction

Dental implants are the preferred fixed oral rehabilitation for replacing lost teeth. With their increasing popularity, peri-implant tissue diseases have also increased, with an estimated prevalence of 19.5% [1]. Peri-implantitis is a condition of the tissues surrounding dental implants characterised by an inflammatory response of the peri-implant connective tissue caused by dysbiotic plaque. This results in progressive loss of supporting bone around the implants [2]. Peri-implantitis has been associated with the formation of a bacterial biofilm on the implant surface consisting mainly of Gram-negative anaerobic bacteria, but opportunistic pathogenic microorganisms such as *P. aeruginosa*, *S. aureus and C. albicans* can also colonise the implant surface and lead to implant failure. [3,4,5].

Undiagnosed and untreated peri-implantitis has an irregular clinical course and rate of progression [6]. The absence of the periodontal ligament, supracrestal attachment fibres, or root cementum on implants distinguishes peri-implantitis from periodontitis [2]. However, they do share the presence of dental plaque around the surface, which is responsible for the immune response and the inflammatory response. For this reason, current therapies focus on decontaminating the implant surface. There are two types of approaches to the treatment of peri-implantitis: surgical (resective and regenerative surgery) and non-surgical (mechanical, chemical, antibiotic, laser methods and oral hygiene education) [4]. Non-surgical therapy should always precede surgical therapy to reduce peri-implant tissue inflammation, assess response to antimicrobial therapy, and ensure effective patient hygiene [7,8]. However, non-surgical therapy does not usually resolve peri-implantitis [9]. Surgical treatment of peri-implantitis includes mechanical decontamination of the implant surface by debridement of the bacterial biofilm and removal of excess cement and inflamed tissue, creation of peri-implant anatomy that promotes its decontamination, and, if possible, regeneration of infra-osseous defects [9,10]. The ideal therapeutic approach must focus on both the removal of pathogens and the prevention of recolonisation of the implant surface [11]. However, there is still no method for decontaminating the surface of titanium implants that can be considered better and more effective than other existing methods [12].

Among the peri-implant therapeutic approaches, the most commonly used can be categorised into four groups: mechanical, chemical, antibiotic and laser [4,13]. A number of methods, such as titanium brushing, particle blasting and, more recently, the electrolytic approach, have shown good results in the decontamination of titanium surfaces, with each method having a different effect on the surface structure [13,14].

The aim of this study was to compare the plaque removal efficacy of an electrolytic decontamination method with three mechanical methods and to evaluate their impact on the implant surface. Our main hypotheses were that the electrolytic treatment would outperform the mechanical treatments in plaque removal and that the mechanical interventions would have a greater effect on the surface characteristics of the dental implants than the electrolytic treatment. *P. aeruginosa* was used as a model plaque bacterium because it adheres to the surface of titanium implants and is often associated with peri-implantitis [5].

## 2. Materials and Methods

### 2.1. Study Design

This in vitro pilot study compared the removal of *P. aeruginosa* plaque from SLA implants (NeoCMIImplant-EB-II active Fixture) using an electrolytic approach (Galvosurge^®^; Galvosurge, Widnau, Switzerland), an erythritol jet system (PerioFlow^®^; EMS, Nyon, Switzerland) and two titanium brushes (R-Brush™ and i-Brush™; NeoBiotech, Seoul, Republic of Korea). The negative control group received no treatment. We focused on SLA implants because of their widespread clinical use worldwide [15].

### 2.2. Sample Size Calculation and Allocation of Interventions

In the absence of studies comparing defined surface cleaning of contaminated implants, we defined a pilot sample of 4 implants per group for a total of 20 implants. 

After being placed in an in vitro peri-implantitis replication model and contaminated with *P. aeruginosa* (Section 2.3), the implants were randomly and equally allocated to each group: C (no treatment); EL (electrolytic method); EJ (erythritol jet system); IB (i-Brush); and RB (R-Brush).

### 2.3. In Vitro Model of Peri-Implantitis and Surface Contamination with P. aeruginosa

SLA implants were immobilised in acrylic models with a tapered defect in the centre and an access hole for implant insertion, leaving the coronal surface exposed (Figure 1).

A *Pseudomonas aeruginosa* PAO1 inoculum was then prepared. Briefly, fresh colonies grown for 24 h on Mueller-Hinton agar (HiMedia Laboratories, India) were used to prepare a bacterial suspension in BHI medium (Brain Heart Infusion; VWR, Belgium) with an optical density at 600 nm (OD600 nm) of 0.1, corresponding to approximately 10^8^ CFU/mL. A further 1:100 dilution was prepared and used to inoculate (1.5 mL) the wells of 12-well plates (JetBiofil Europe, Spain) already containing the acrylics in the inverted position (implants down), as shown in Figure 1. The plates were incubated at 37 °C for 24 h to allow biofilm formation on the implants. The acrylics were then removed and transferred in the same position to a new well containing 2 mL of PBS 1× (Phosphate-Buffered Saline; Fisher Scientific, Oxford, UK) so that the biofilms formed on the surface of the implants were washed once. The acrylic was then positioned with the implants side-up and ready for the subsequent specific treatment.

### 2.4. Plaque Removal Interventions

Electrolytic decontamination [Figure 2a]: Electrolytic decontamination device using galvanic currents (Galvosurge; Galvosurge Dental AG, Widnau, Switzerland), used according to the manufacturer’s instructions. The head of the device is inserted into the implant and held in place throughout the cleaning process. The Galvosurge cleaning solution is pumped in, and a galvanic current forms hydrogen bubbles on the implant surface to release the adherent biofilm. The unit automatically stops working when it reaches 100% (2 min).Air abrasion with erythritol powder [Figure 2b]: An erythritol-based air-powder abrasive device with a subgingival plastic nozzle (Perioflow; EMS, Nyon, Switzerland), used according to the manufacturer’s recommendations, with the “liquid” and “power” settings set to the maximum (2 min).Mechanical debridement with i-Brush™ [Figure 2c]: Titanium brushes with stainless bristles (i-Brush; NeoBiotech, Seoul, Republic of Korea) used with irrigation at 10,000 rpm (2 min).Mechanical debridement using R-Brush™ [Figure 2d]: Rotary titanium brushes (5.6/6.6 in diameter; R-Brush; NeoBiotech, Seoul, Republic of Korea) were used at a rotational speed of 5000 rpm under irrigation with water (2 min).

### 2.5. Biofilm Quantification

Each implant, either untreated (control) or treated, was carefully removed from the acrylic and placed in the well of a new 12-well plate containing 2 mL PBS 1×. Each 12-well plate was sealed with parafilm and placed in an ultrasonic bath (TPC Dentsonic UC-1000 10L Ultrasonic Cleaner) for 20 min to dislodge and disperse the biofilm from the implants. Then, from each well containing the dispersed biofilm in PBS 1×, 100 μL was used (after vigorous up-down pipetting for better homogenisation) to perform serial 1:10 dilutions. Controls were diluted from 10^−1^ to 10^−6,^ and treated conditions were diluted from 10^−1^ to 10^−4^. Then 100 μL of the last four dilutions (10^−3^ to 10^−6^ for the control conditions and 10^−1^ to 10^−4^ for the treated conditions) were plated on Mueller-Hinton agar plates and incubated overnight at 37 °C. The next day, colony-forming units (CFU) were counted on agar plates containing between 30 and 300 colonies. CFU/cm^2^ values were then calculated for each condition. The area (cm^2^) of the implant where the biofilm was formed was calculated using the following formula: *A* = 2.π.r.h, where ‘r’ is the radius of the implant and ‘h’ is the height immersed in the inoculum. CFU/cm^2^ were converted to log_10_ CFU/cm^2,^ and the mean and standard deviation were calculated for each study group.

### 2.6. Scanning Electron Microscopy

The samples were observed using the following parameters: 350× magnification, 40 mm working distance (WD), and 30 kV high voltage. Images were processed using the GNU Image Manipulation Program—gimp-2.10.

### 2.7. Statistical Analysis 

Differences in bacterial colony counts (log_10_ CFU/cm^2^) between control and treatment groups were evaluated by one-way ANOVA with Tukey’s multiple comparison test. Hypothesis tests were two-tailed, and values of *p* < 0.05 were considered significant.

## 3. Results

### 3.1. Quantitative Analysis of Implant-Adhered Bacteria

With the exception of the R-Brush treatment, the number of *P. aeruginosa* cells removed from the SLA implants after the different types of treatment was significantly lower compared to the control implants (no treatment) (Figure 3, Table 1 and Table 2). Consistent with this, the RB group had the highest mean log_10_ CFU/cm^2^ of all treatment groups. On the other hand, the EJ (erythritol jet) group had the lowest mean CFU/cm^2^ values (Figure 3 and Table 1). These data suggest that air polishing with an erythritol jet is the best method for decontaminating implants and that the titanium R-Brush treatment is not effective for this purpose.

### 3.2. Qualitative Evaluation of Treated Implants by Scanning Electron Microscopy

The surface of the four implants in each group was evaluated by scanning electron microscopy, and the valley area between the coils was observed in two different areas of each implant, giving a total of eight images per group (Figure 4). There was a noticeable change in the surface of the implants treated with i-Brush™ and R-Brush™. In addition, when comparing these two brushes, there appears to be a greater change on the surfaces instrumented with i-Brush™. In the EL and EJ groups, no significant changes were observed on the different surfaces compared to the control group.

## 4. Discussion

This study showed that electrolytic decontamination, erythritol jet and i-Brush titanium brushes significantly reduced the bacterial load on titanium implant surfaces. However, none of the methods achieved complete decontamination of the implants. Lower bacterial counts (log10 CFU/cm^2^) were observed on implants treated with the air polishing system, and higher bacterial counts were observed on implants treated with R-Brush, but the differences between methods did not reach statistical significance, indicating that all decontamination strategies except R-Brush performed similarly. As a result, the hypothesis of this study that the electrolytic treatment would be better than the mechanical treatments in biofilm removal is rejected. To our knowledge, there are no in vitro studies performed under the same conditions comparing the effectiveness of the decontamination methods evaluated in this study.

SEM is commonly used for qualitative analysis of implant surfaces because of its ability to provide information about the surface topography of the samples [11,13,16,17,18,19,20,21,22,23,24,25]. No significant changes were observed on the implant surfaces in the EL and EJ groups, as observed by SEM. In contrast, implants instrumented with the titanium brushes (groups IB and RB) showed significant surface changes in the form of scratches compatible with the brush bristles. These results confirm the second hypothesis of this study that mechanical interventions have a greater effect on the surface characteristics of dental implants than electrolytic treatment.

Mechanical debridement of biofilm using titanium brushes has shown some efficacy in previous studies, i.e., a reasonable reduction in the bacterial count of the surface under investigation [17,18,19,25,26]. However, brushing did not show superior efficacy compared to other existing methods, and the authors also reported significant changes in the titanium surfaces caused by brushing, using qualitative analysis by SEM. Our results confirm the previously mentioned studies, i.e., the i-Brush showed some decontamination efficacy without evidence of superiority compared to the other methods used and caused remarkable changes in the surface of the implants. The relatively poor decontamination performance of the R-Brush observed in the current study is likely related to its low rotation speed (5000 rpm, half that of the i-Brush).

The particle jet system is a widely studied method, especially when using sodium bicarbonate particles [22,24,27,28] and glycine [13,14,21,22,23,25,27,28,29,30]. It has been found that when smaller particle sizes such as glycine (25 μm) or erythritol (14 μm) are used, the surface changes are not significant compared to sodium bicarbonate (40-65 μm), which has larger particle sizes and consequently higher abrasive power [22]. In the present study, we found that the erythritol-chlorhexidine particle jet system (14 μm) had no significant effect on the surface of the SLA implants. These results are consistent with previous studies showing that this treatment does not cause significant changes to the surface of implants [22].

Recently, a new electrolytic approach has been developed, which consists of applying a galvanic current to the implant, resulting in the formation of hydrogen bubbles on its surface, thus releasing the adherent biofilm [13]. Electrochemical treatment of dental implants combines the antibacterial action of bactericides with the direct oxidation of bacterial enzymes and proteins, resulting in a low free concentration of harmful chemicals, allowing the surface of an implant to be cleaned without altering its microtopography and without affecting its physical properties [13,31,32,33]. In these studies, electrolytic decontamination outperformed glycine jetting, and only electrolytic decontamination was able to completely decontaminate the implant surfaces [13]. In the present study, electrolytic decontamination did not show high decontamination performance, with higher CFU/cm^2^ values than the erythritol jet method. The difference in results between the present study and previous studies may be due to the use of different particles in the jetting system and the difference in the bacterial system used, i.e., *P. aeruginosa* biofilm vs saliva-derived biofilm. The bacteria present in the saliva-derived biofilm were not characterised, and the salivary microbiome is usually highly heterogeneous [34], making it impossible to compare the results. However, we have used a robust and strongly adherent biofilm model of *P. aeruginosa*, which can be considered more difficult to remove, thus contributing to the poorer results obtained in this study [35]. *P. aeruginosa* is a gram-negative bacterium that grows easily under a wide range of conditions and temperatures and has a high ability to adhere to surfaces [35]. In addition, *P. aeruginosa* is commonly found on dental implants in patients with peri-implantitis [5]. For these reasons, *P. aeruginosa* was chosen as the model bacterium for this research. On the other hand, the SEM evaluation of the surface of the implants treated by the electrolytic decontamination method confirms previous results [13,33], i.e., the electrolytic decontamination system does not cause significant changes in the surface of the implants.

This in vitro study has limitations such as the use of implants with different geometries, different areas exposed to contamination and decontamination, non-uniformity of the biofilm formed on the implant surface due to the characteristics of *P. aeruginosa* biofilms, and the use of monospecies biofilms. Studies using multi-species biofilms, which better mimic the complex biofilm found in the oral cavity, are needed to better determine the usefulness of the methods analysed in this study for implant decontamination.

## 5. Conclusions

This pilot study suggests that electrolytic treatment, erythritol-chlorhexidine particle jetting and i-Brush™ brushing have similar performance in removing *P. aeruginosa* biofilm from dental implants. Further studies are needed to evaluate the removal of more complex biofilms, which represent dental plaque. Titanium brushes caused significant changes to the implant surface, the effects of which need to be evaluated.

## Figures and Tables

**Figure 1 jcm-12-01703-f001:**
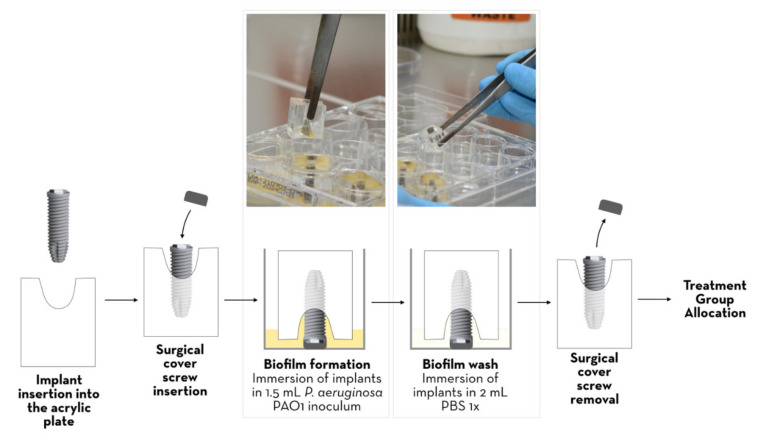
In vitro model of peri-implantitis and surface contamination with *P. aeruginosa*. After implant placement in the acrylic plate, the surgical cover screw was inserted. The implants were then inverted and immersed in a *P. aeruginosa* inoculum and incubated at 37 °C for 24 h to allow biofilm formation. The biofilms formed on the implants were then washed by immersion in 2 mL PBS 1×. Surgical cover screws were removed prior to treatment group allocation.

**Figure 2 jcm-12-01703-f002:**
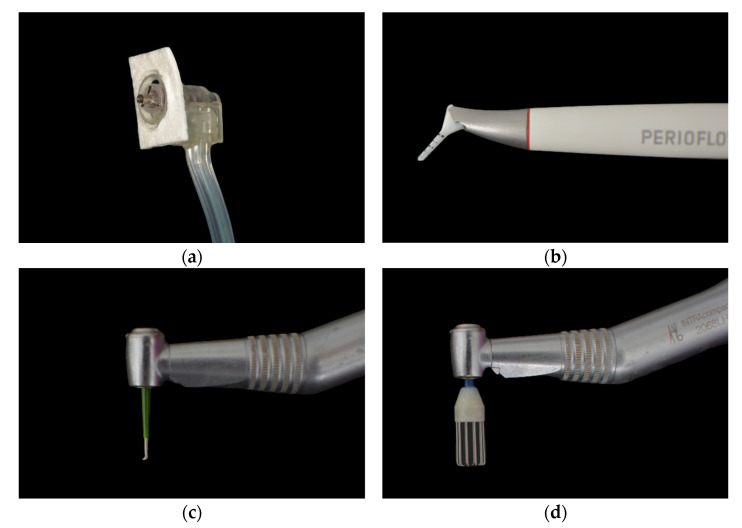
Instruments used to remove plaque from the implants: (**a**) Galvosurge^®^; (**b**) Perioflow^®^; (**c**) i-Brush^™^; (**d**) R-Brush^™^.

**Figure 3 jcm-12-01703-f003:**
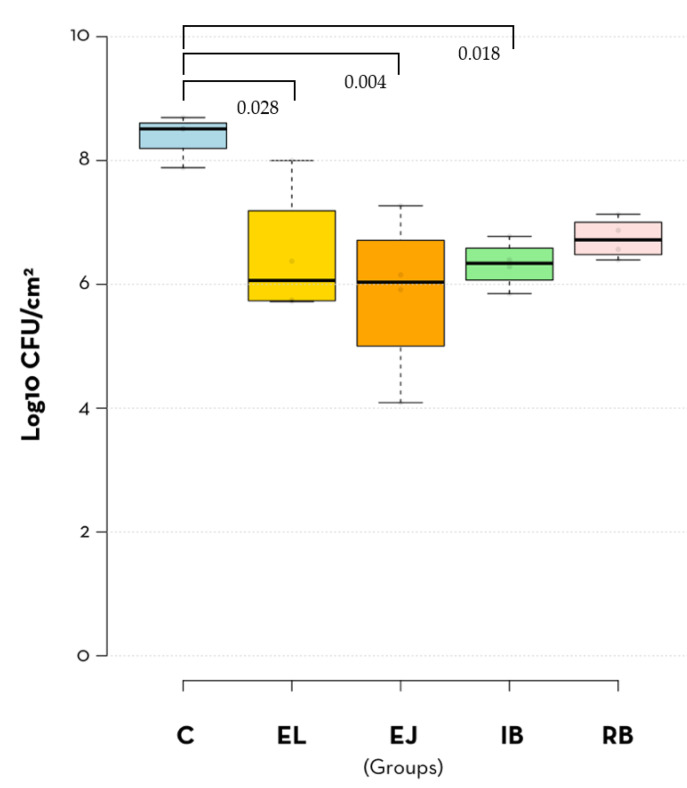
*P. aeruginosa* PAO1 CFU counts (log10 CFU/cm^2^) obtained—from the no treatment control group (C) and for the treatment groups (electrolytic decontamination (EL); erythritol jet (EJ); i-Brush™ (IB) and R-Brush™ (RB)). Mean and standard deviation are presented.

**Figure 4 jcm-12-01703-f004:**
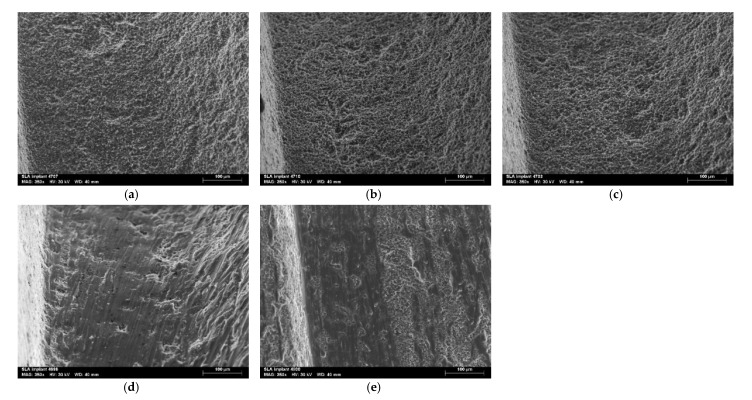
Surface analysis of treated implants by scanning electron microscopy: (**a**) control group; (**b**) EL: electrolytic decontamination; (**c**) EJ: erythritol jet; (**d**) IB: i-Brush™; (**e**) RB: R-Brush™. Representative images are shown.

**Table 1 jcm-12-01703-t001:** Descriptive results (log_10_ CFU/cm^2^) for each group.

	PC	EL	EJ	IB	RB
Mean (SD)	8.39 (0.36)	6.46 (1.07)	5. 86 (1.32)	6.32 (0.38)	6.74 (0.33)
SE	0.18	0.54	0.66	0.19	0.16
95% CI	7.83–8.96	4.76–8.16	3.76–7.95	5.72–6.93	6.22–7.26
Minimum	7.88	5.72	4.09	5.85	6.39
Maximum	8.69	8.00	7.27	6.77	7.13

**Table 2 jcm-12-01703-t002:** Multiple comparisons between the groups (p values are shown).

	C	EL	EJ	IB	RB
C	-	0.028	0.004	0.018	0.070
EL	-	-	0.824	0.999	0.987
EJ	-	-	-	0.920	0.549
IB	-	-	-	-	0.947
RB	-	-	-	-	-

## Data Availability

Not applicable.

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
