# Peer review of "Dental Implant Surface Decontamination and Surface Change of an Electrolytic Method versus Mechanical Approaches: A Pilot In Vitro Study"

_jcm, 2023, doi:10.3390/jcm12041703_

Round 1
Reviewer 1 Report
This study wants to compare the special bacteria removal efficacy and possible damage on implant surface by different clinical methods. The results are relatively simple and lack of meticulous discussion.
1. Does P. aeruginosa is the main kind bacteria of peri-implant diseases, and why not to compare more kinds of identified monospecies biofilms, such as P. gingivalis, Staph. aureus, Staph. anaerobius, T. forsythia and etc.
2. How to determine the treatment time for each group by different technique, and there is no statement in methods. Is it possible to affect the results.
3. Titanium brushes caused significant changes to the implant surface, but there is significant difference on bacteria removal by i-brush or r-brush. We do not find any explanation on it and this may be an interesting aspect for more analysis.
Author Response
Dear Editorial Board,
We appreciate the opportunity to revise and resubmit our manuscript "Dental implant surface decontamination and surface change of an electrolytic method versus mechanical approaches: a pilot in vitro study" (Manuscript ID jcm-2228471).
We thank the editor and reviewers for their comments, all of which have been considered and addressed.
Changes to the manuscript are highlighted in the revised manuscript. Our point-by-point responses to all comments are detailed below. We are happy to consider further revisions and thank you for your interest in our research.
REVIEWER 1:
This study wants to compare the special bacteria removal efficacy and possible damage on implant surface by different clinical methods. The results are relatively simple and lack of meticulous discussion.
1. Does P. aeruginosa is the main kind bacteria of peri-implant diseases, and why not to compare more kinds of identified monospecies biofilms, such as P. gingivalis, Staph. aureus, Staph. anaerobius, T. forsythia and etc.
Our response: P. aeruginosa is an important opportunistic bacterium that, as mentioned in the introduction and discussion of the revised paper, is increasingly found on dental implants in peri-implantitis patients. It is a good model for this type of plaque removal study because it forms a strongly adherent biofilm on dental implants. We could have used other species for contamination of the dental implants, as the reviewer rightly suggests. However, the species mentioned by the reviewer lack the ability to adhere to dental implants and/or require different and more selective growth conditions compared to P. aeruginosa, including the use of anaerobic conditions. For a pilot study, we believe that the model bacterium and conditions used were adequate. In the future, we plan to conduct studies using anaerobic bacteria.
2. How to determine the treatment time for each group by different technique, and there is no statement in methods. Is it possible to affect the results.
Our response: The treatment time was always the same as we followed the GalvoSurge recommended average time of 2 minutes. We have standardised the remaining treatment protocols according to this reference time. The treatment time is now reported in the revised manuscript.
3. Titanium brushes caused significant changes to the implant surface, but there is significant difference on bacteria removal by i-brush or r-brush. We do not find any explanation on it and this may be an interesting aspect for more analysis.
Our response: The relatively poor decontamination performance of the R-Brush observed in our study is likely related to its low rotation speed (5,000 rpm, half that of the i-Brush). We have added this sentence to the discussion.
Reviewer 2 Report
Abstract:
- Line 19-20 ''We also evaluated changes in the implant surface after each approach.'' Avoid use ''we'' in a scientific article. Reformulate example, Changes in the implant surface were then evaluated after each approach.
Introduction:
- It is a little short. I suggest you to add some sentences and references on the state of the art focusing on the currently described results of the alternatives you described. Example, surgical procedures are usually invasive etc... non surgical procedures provided some positive results but are not always predictable etc... this will provide the state of the art and it will help to understand the reason of testing a novel approach.
- Please move the study hypothesis from the M&M section to the end of the introduction after the study aim
Discussion:
- Please add if the hypothesis were accepted or rejected based on the results of the study.
- Add an interpretation on why the complete removal of the biofilm was not achieved (example singular application paving the way for further study)
- Add why the study is considered a pilot study (are you planning to perform more samples?)
- A recent article on similar topic was published describing a novel methods for the decontamination of the titanium surface. I encourage you to add it and cite it in the text to help the reference improvement with recent articles on the topic
Alovisi, M.; Carossa, M.; Mandras, N.; Roana, J.; Costalonga, M.; Cavallo, L.; Pira, E.; Putzu, M.G.; Bosio, D.; Roato, I.; Mussano, F.; Scotti, N. Disinfection and Biocompatibility of Titanium Surfaces Treated with Glycine Powder Airflow and Triple Antibiotic Mixture: An In Vitro Study. Materials 2022, 15, 4850. https://doi.org/10.3390/ma15144850
Conclusion:
''This pilot study suggests that electrolytic, erythritol-chlorhexidine particle jetting 242 and i-BrushTM brushing are equally effective methods for removing P. aeruginosa biofilm from dental implants.'' Based on the results of the study, the complete elimination of the biofilm was not achieved for any of the tested methods. Therefore it is a mistake to state that they are effective. Please reformulate the conclusion in both the abstract and main text according to the results.
Author Response
Dear Editorial Board,
We appreciate the opportunity to revise and resubmit our manuscript "Dental implant surface decontamination and surface change of an electrolytic method versus mechanical approaches: a pilot in vitro study" (Manuscript ID jcm-2228471).
We thank the editor and reviewers for their comments, all of which have been considered and addressed.
Changes to the manuscript are highlighted in the revised manuscript. Our point-by-point responses to all comments are detailed below. We are happy to consider further revisions and thank you for your interest in our research.
REVIEWER 2:
Abstract:
- Line 19-20 ''We also evaluated changes in the implant surface after each approach.'' Avoid use ''we'' in a scientific article. Reformulate example, Changes in the implant surface were then evaluated after each approach.
Our response: We have reworded this to read as follows: “Changes in the implant surface after each approach were also evaluated”.
Introduction:
- It is a little short. I suggest you to add some sentences and references on the state of the art focusing on the currently described results of the alternatives you described. Example, surgical procedures are usually invasive etc... non surgical procedures provided some positive results but are not always predictable etc... this will provide the state of the art and it will help to understand the reason of testing a novel approach.
Our response: The introduction was expanded as rightly suggested by the reviewer.
- Please move the study hypothesis from the M&M section to the end of the introduction after the study aim.
Our response: We have moved the study hypothesis from the M&M section to the end of the introduction, after the study aim, as recommended.
Discussion:
- Please add if the hypothesis were accepted or rejected based on the results of the study.
Our response: Done as suggested
- Add an interpretation on why the complete removal of the biofilm was not achieved (example singular application paving the way for further study)
Our response: In lines 253-267 of the discussion of the revised manuscript we offer a reasonable explanation for the incomplete removal of the P. aeruginosa biofilm.
- Add why the study is considered a pilot study (are you planning to perform more samples?)
Our response: Because of its design, this study is a pilot study. In fact, we are planning a study with a much larger sample size and with additional bacterial species.
- A recent article on similar topic was published describing a novel methods for the decontamination of the titanium surface. I encourage you to add it and cite it in the text to help the reference improvement with recent articles on the topic
Alovisi, M.; Carossa, M.; Mandras, N.; Roana, J.; Costalonga, M.; Cavallo, L.; Pira, E.; Putzu, M.G.; Bosio, D.; Roato, I.; Mussano, F.; Scotti, N. Disinfection and Biocompatibility of Titanium Surfaces Treated with Glycine Powder Airflow and Triple Antibiotic Mixture: An In Vitro Study. Materials 2022, 15, 4850. https://doi.org/10.3390/ma15144850
Our response: We thank the reviewer for the suggestion. We added a mention to this important paper in the introduction.
Conclusion:
''This pilot study suggests that electrolytic, erythritol-chlorhexidine particle jetting 242 and i-BrushTM brushing are equally effective methods for removing P. aeruginosa biofilm from dental implants.'' Based on the results of the study, the complete elimination of the biofilm was not achieved for any of the tested methods. Therefore it is a mistake to state that they are effective. Please reformulate the conclusion in both the abstract and main text according to the results.
Our response: We agree with this comment and have rephrased it to "have similar performance in" instead of "are equally effective in".
Round 2
Reviewer 1 Report
Electrolytic decontamination is distinguished from other mechanical techniques, it based on different mechanism. It is better to add more discussion on this aspect and extend out further research and clinical suggestion.
Author Response
REVIEWER 1:
Electrolytic decontamination is distinguished from other mechanical techniques, it based on different mechanism. It is better to add more discussion on this aspect and extend out further research and clinical suggestion.
Our response: To accommodate this suggestion, we have added a new paragraph on lines 241-247 of the Discussion.
Reviewer 2 Report
Dear Authors,
thank you for addressing all my comments. I don't have any further revisions to require.
Author Response
Thank you for your time and effort reviewing our manuscript.